# Oncological Outcomes in Patients with Metastatic Urothelial Carcinoma after Discontinuing Pembrolizumab as a Second-Line Treatment: A Retrospective Multicenter Real-World Cohort Study

**DOI:** 10.3390/biomedicines10092243

**Published:** 2022-09-09

**Authors:** Toyohiro Yamada, Keita Nakane, Torai Enomoto, Masayuki Tomioka, Tomoki Taniguchi, Takashi Ishida, Kaori Ozawa, Kimiaki Takagi, Hiroki Ito, Shinichi Takeuchi, Makoto Kawase, Kota Kawase, Daiki Kato, Manabu Takai, Koji Iinuma, Shigeaki Yokoi, Masahiro Nakano, Takuya Koie

**Affiliations:** 1Department of Urology, Gifu University Graduate School of Medicine, Gifu 5011194, Japan; 2Department of Urology, Matsunami General Hospital, Hashima-gun 5016062, Japan; 3Department of Urology, Japanese Red Cross Takayama Hospital, Takayama 5068550, Japan; 4Department of Urology, Chuno Kosei Hospital, Seki 5013802, Japan; 5Department of Urology, Ogaki Municipal Hospital, Ogaki 5038502, Japan; 6Department of Urology, Gifu Municipal Hospital, Gifu 5008513, Japan; 7Department of Urology, Daiyukai Daiichi Hospital, Ichinomiya 4918551, Japan; 8Department of Urology, Toyota Memorial Hospital, Toyota 4718513, Japan; 9Department of Urology, Central Japan International Medical Center, Minokamo 5058510, Japan; 10Department of Urology, Gifu Prefectural General Medical Center, Gifu 5008717, Japan

**Keywords:** multicenter cohort study, overall survival, oncological outcomes after pembrolizumab, metastatic urothelial carcinoma

## Abstract

The treatment options are currently limited, and the oncological outcomes remain unclear, for patients with metastatic urothelial carcinoma (mUC) with or without third-line systemic therapy. We aimed to evaluate the oncological outcomes in real-world daily clinical practice after platinum-based chemotherapy followed by pembrolizumab for mUC. This retrospective, multicenter cohort study included patients with mUC who received second-line pembrolizumab in Japan. The patients were divided into the treatment group (those who received third-line treatment) and the BSC group (those who did not receive other treatments). The primary endpoint of this study was to evaluate the oncological outcomes. Of 126 patients enrolled in this study, 40 received third-line therapy. The median follow-up period was 8.0 months. The median overall survival (OS) times were nine months in the BSC group and 17 months in the treatment group (*p* < 0.001). The median progression-free survival (PFS) times were 4 months in the BSC group and 14 months in the treatment group (*p* < 0.001). In the multivariate analysis, performance status and liver metastasis were significantly associated with OS. Third-line therapy may have clinical potential advantages for improving the oncological outcomes in patients with mUC.

## 1. Introduction

Urothelial carcinoma (UC) is the most common histological type of urinary tract cancer [1]. Although most patients with UC were diagnosed with a non-muscle invasive tumor, 5% had metastatic UC (mUC) at diagnosis [2]. Generally, mUC has been recognized as an incurable and highly lethal disease with a 5-year overall survival (OS) of only 4.6% [3]. Therefore, platinum-based combination chemotherapies as first-line treatment options and immune-oncology (IO) agents targeting the programmed death-receptor 1 or programmed death-ligand 1 (PD-1/L1) as second-line treatment options are recommended for mUC according to several guidelines [4,5,6].

The OS in patients with mUC who received first-line platinum-based chemotherapy ranged from 9 to 15 months [7,8]. Additionally, the median OS of mUC patients who received pembrolizumab as a second-line treatment was approximately 10 months although the IO treatment had a more acceptable level of adverse events and was associated with a longer duration of response compared with chemotherapy [9,10]. Recently, results from the EV-301 trial showed that the median OS and progression-free survival (PFS) associated with enfortumab vedotin (EV) were significantly longer than those associated with antitumor agents in patients with mUC who received platinum-based chemotherapy followed by PD-1/L1 inhibitors and developed disease progression during or after medication treatment [11]. Therefore, EV will become the mainstream third-line therapy for mUC. However, after the discontinuation of PD-1/L1 inhibitor therapy, all patients with mUC cannot receive subsequent systemic therapy [12]. Additionally, the current treatment options have been limited, and the oncological outcomes, especially OS, remain unclear for patients with mUC who received versus did not receive third-line systemic therapy [12,13,14].

Hence, we aimed to evaluate the oncological outcomes in real-world daily clinical practice after platinum-based chemotherapy followed by pembrolizumab treatment for mUC.

## 2. Materials and Methods

### 2.1. Patients

This study was approved by the Institutional Review Board of Gifu University (authorization number: 2021-B080). The requirement for obtaining informed consent was waived due to the retrospective nature of the study. The provisions of the ethics committee and ethics guidelines in Japan did not require written consent because the study information was disclosed to the public, as is the case with retrospective and observational studies that use materials such as existing documentation. The details of this study can be accessed at https://www.med.gifu-u.ac.jp/visitors/disclosure/docs/2021-B080.pdf (accessed on 15 May 2022).

This retrospective, multicenter cohort study included patients with mUC after receiving second-line pembrolizumab at 10 institutions in Japan between December 2017 and August 2021. All enrolled patients had histologically confirmed UC with distant metastases and had received platinum-based combination chemotherapy followed by pembrolizumab treatment. None of the enrolled patients received immunotherapy as first-line therapy. We excluded patients whose treatment response was not evaluated after pembrolizumab initiation and those with missing data. The clinicopathological and laboratory parameters included patient’s age, sex, height, weight, body mass index (BMI), Eastern Cooperative Oncology Group performance status (ECOG-PS) [15], smoking history, primary tumor site, metastatic sites, definitive therapy for primary tumor, hemoglobin level (Hb), serum albumin (Alb) level, c-reactive protein (CRP), and neutrophil-to-lymphocyte ratio (NLR). All tumors were staged according to the 8th Edition American Joint Committee on Cancer Staging Manual [16].

### 2.2. Treatment Schedule

All participants received platinum-based chemotherapy as first-line treatment, and pembrolizumab was subsequently administered as second-line therapy until disease progression was detected on radiographic examination, the patient refused treatment, or intolerance developed in the form of treatment-related AEs according to the National Cancer Institute Common Terminology Criteria for Adverse Events (version 5.0) [17]. In the KEYNOTE045 trial, pembrolizumab was administered at a dosage of 200 mg every three weeks. Therefore, in Japan, the treatment dose of pembrolizumab is 200 mg every three weeks, regardless of the BMI of the patient; the enrolled patients in this study also received 200 mg pembrolizumab every three weeks.

The patients were divided into the treatment group (those who received third-line treatment) and BSC group (those who did not receive other treatments).

### 2.3. Patient Evaluation

The patients’ baseline characteristics were obtained through complete history taking; physical examination; and chest, abdominal, and pelvic computed tomography (CT) examinations. All patients underwent CT every 2 months until disease progression according to the results of radiological evaluation. The best overall response (BOR) after pembrolizumab therapy was documented as complete response (CR), partial response (PR), stable disease (SD), or progressive disease (PD) using the Response Evaluation Criteria in Solid Tumors (RECIST) guidelines, version 1.1 [18]. The cutoff points for age, BMI, Hb, Alb, CRP, and NLR were used as the median values. In addition, the patients were divided into two groups according to the BOR: those who achieved CR or PR (responder group) and those who had SD or PD (nonresponder group).

### 2.4. Endpoints and Statistical Analysis

The primary endpoint of this study was to evaluate the oncological outcomes, including OS and PFS. The secondary endpoint was determining the association between clinicopathological features and oncological outcomes. The data were analyzed using JMP 14 software (SAS Institute Inc., Cary, NC, USA). The date of pembrolizumab administration was used as the starting point for estimating the OS and PFS. OS was defined as the time from the initiation of pembrolizumab treatment to death from any cause. PFS was defined as the time from the initiation of pembrolizumab treatment to disease progression. The Kaplan–Meier method was used to evaluate the OS and PFS, and the differences were assessed according to the clinical variables using the log-rank test. Multivariate analysis was performed using the Cox proportional hazards model. All *p* values were two sided, and a *p* value of <0.05 was considered significant.

## 3. Results

### 3.1. Patient Characteristics

A total of 126 patients were enrolled in this study, and 40 patients received the following treatments as third-line therapy: gemcitabine-based combination chemotherapy in 22 patients, radiation or surgical treatment for metastatic sites in 3, gemcitabine monotherapy in 1, docetaxel monotherapy in 1, and other anti-cancer therapies in 10.

The demographic data on patients who received and did not receive third-line therapies are listed in Table 1. The patients’ median age and BMI were 72 years (interquartile range [IQR], 69–78 years) and 21.9 kg/m^2^ (IQR, 19.3–24.2 kg/m^2^), respectively. In this cohort, 73.0% of the patients were men, 55.6% had a smoking history, and 19.0% had an ECOG-PS score of ≥2. The most common primary lesions and metastatic sites were the urinary bladder (57.9%) and lymph nodes (75.4%). Approximately 84.9% of the patients underwent definitive therapy for primary lesions.

The blood biochemical findings before and after pembrolizumab therapy are shown in Table 2. Although the CRP, Alb, Hb, and NLR levels were normal in all patients before and after pembrolizumab therapy, the CRP and NLR levels in the BSC group were significantly higher before and after pembrolizumab administration than those in the treatment group. Conversely, the Hb and Alb levels in the treatment group were significantly higher before and after pembrolizumab administration than those in the BSC group.

### 3.2. Oncological Outcomes

The median follow-up was 8.0 months (IQR, 3.0–15.0 months). At the end of follow-up, 80 patients (64.3%) had died from UC.

The median OS times were 17 months (95% CI, 12 months–not applicable) in the treatment group and 9 months (95% confidence interval [CI], 5–14 months) in the BSC group (*p* < 0.001; Figure 1). The median PFS times were 14 months (95% CI, 8–24 months) in the treatment group and 4 months (95%CI, 2–6 months) in the BSC group (*p* < 0.001; Figure 2).

According to the BOR after pembrolizumab administration, the median OS times were 5 months (95% Cl, 4–8 months) in the nonresponder group and 17 months (95% CI, 12–not applicable) in the responder group (*p* = 0.001; Figure 3A). In addition, the enrolled patients were divided into four groups: patients who received and did not receive various forms of treatments were evenly distributed in the responder (groups 1 and 2, respectively) and nonresponder groups (groups 3 and 4, respectively). The median OS was not reached in group 1 and OS was 17 months in group 2, 9 months in group 3, and 3 months in group 4 (Figure 3B). Patients in the responder group who received third-line treatment had a significantly longer OS than those in the nonresponder group who did not receive third-line treatment (*p* = 0.004). In the nonresponder group, patients who received various forms of treatments had significantly longer OS than those who did not receive any form of treatment (*p* = 0.011).

Although the treatment group had significantly longer OS compared with that in the BSC group as shown in the univariate analysis, ECOG-PS and liver metastasis were significantly associated with OS in the multivariate analysis (Table 3).

## 4. Discussion

In general, mUC is recognized as an incurable disease with a poor prognosis [2,3,18]. According to several guidelines, treatment strategies should be developed using sequential systemic therapies, including platinum-based chemotherapy, PD-1/L1 inhibitor therapy, and other anticancer agents, to improve the oncological outcomes in patients with mUC [4,5,6]. In real-world clinical practice, however, patients with mUC who previously received platinum-based chemotherapy and PD-1/L1 inhibitor therapy showed limited response to subsequent therapies and shorter OS and PFS [18]. In the KEYNOTE045 trial, Bellmunt et al. demonstrated the efficacy of pembrolizumab in patients with advanced UC who experienced disease relapse or progression after first-line platinum-based chemotherapy [9]. Although oncological outcomes in patients with advanced UC or mUC receiving pembrolizumab as second-line therapy were evaluated in the trial, the efficacy of third-line therapy after pembrolizumab discontinuation was not investigated [9]. Therefore, we focused on patients with mUC who discontinued pembrolizumab in our study.

Taxane, alone or in combination with a variety of anticancer agents, is commonly used in the treatment of mUC that progressed after treatment with platinum-based chemotherapy and PD-1/L1 inhibitor therapy despite the reports of a few studies supporting their use [14,18,19]. Based on a retrospective study that reviewed 8 trials including 370 patients, the combination of taxane with other chemotherapeutic agents improved the OS (hazard ratio [HR], 0.60; *p* = 0.001) and PFS (HR, 0.57; *p* < 0.001) [14]. Hepp et al. reported that the median real-world OS times were 7.6 months (95% CI, 5.2–14.4 months) in the taxane monotherapy cohort and 8.9 months (95% CI, 2.4–4.0 months) in the any therapy cohort [18]. The median PFS times were 2.9 months (95% CI, 2.4–4.0 months) in the taxane monotherapy cohort and 3.6 months (95% CI, 2.7–4.7 months) in the any therapy cohort [18]. Due to the decreased survival outcomes in the taxane monotherapy cohort, more patients with aggressive disease were included in this group; in addition, the proportion of patients who previously received taxane monotherapy was higher than that of patients who received other forms of treatment (84.7% had ≥2 previous treatments in the taxane group vs. 76% in the any therapy cohort) [18]. In our study, patients who received third-line therapy had a longer OS (17 months) than those in a previous study [18]. Additionally, a relatively large proportion of patients with a short duration of anticancer agent use might have enrolled in our study owing to early recurrence or metastases after chemotherapy. Therefore, patients who undergo third-line chemotherapy may have relatively improved sensitivity to anticancer drugs. Hence, these patients might achieve improved oncological outcomes after third-line therapy.

Conversely, only a few studies were specifically conducted in patients with mUC who received third-line therapy. Matsumoto et al. reported the efficacy of gemcitabine and nedaplatin therapy in 10 patients with mUC who previously received methotrexate, vinblastine, doxorubicin, and cisplatin, followed by gemcitabine and paclitaxel [20]. The median OS and PFS were 8.8 months and 5.0 months, respectively [20]. Similarly, the median OS and PFS were 6.3 and 4.1 months, respectively, in 23 patients with mUC treated with pegylated liposomal doxorubicin as third-line chemotherapy, whereas the median OS and PFS were 7.3 and 2.0 months, respectively, in 13 patients treated with third-line gemcitabine monotherapy [21,22]. Di Lorenzo et al. reported that cyclophosphamide monotherapy was associated with an OS of 38 weeks and a PFS of 18 weeks, whereas platinum-based combination chemotherapy was associated with an OS of 8 and a PFS of 5 weeks [19]. Although only a small number of patients were enrolled in these studies, they showed the limited use of third-line chemotherapy for mUC in a real-world practice.

As another important issue related to the use of third-line therapy, all patients with mUC cannot receive other forms of aggressive treatments after PD-1/L1 inhibitor therapy. Although there were no data on the precise reasons for discontinuing PD-1/L1 inhibitor therapy, approximately half the patients did not receive subsequent therapy after PD-1/L1 treatment [18]. Other real-world studies have reported that only 25–35% of the patients could receive subsequent systemic treatment following the discontinuation of PD-1/L1 inhibitor therapy [12,23]. In our study, patients who achieved CR or PR after pembrolizumab administration had better oncological outcomes than those with SD or PD. This result suggests that patients with CR or PD might have maintained good ECOG-PS and better OS after receiving third-line chemotherapy.

Recently, the EV-301 trial demonstrated significantly longer oncological outcomes with EV than with investigator-chosen chemotherapy in patients with mUC who previously received platinum-based chemotherapy and PD-1/L1 inhibitor therapy [11]. In our study, the median OS and PFS in the treatment group were slightly longer than the oncological outcomes reported in the EV arm from the EV-301 trial although the enrolled patients in our study were older than those in the EV-301 trial [11]. The reasons why the patients in this study achieved longer oncological outcomes than those in the EV-301 trial remain uncertain; moreover, patients who are appropriate for receiving subsequent treatment after pembrolizumab therapy may experience improvements in their oncological outcomes. Additionally, it may be important to maintain good ECOG-PS, especially physical and mental status, in order to continue receiving systemic therapy in patients with mUC.

There are several limitations to our study. First, this was a retrospective study and was conducted using multicenter data. Therefore, this study had an inherent potential for bias based on diagnostic and therapeutic variations among these institutions. Second, a relatively small number of patients were enrolled in this study, and the follow-up period was relatively short. Additionally, approximately 70% of the patients in this study were unable to receive third-line chemotherapy. Third, this study did not include patients who received chemotherapy using anticancer agents for mUC as a control group. Finally, data on several adverse events associated with third-line treatments were not obtained.

## 5. Conclusions

Third-line therapy may have clinical potential advantages for improving the oncological outcomes in patients with mUC. Additionally, this study indicated that the patients who had better clinical response to pembrolizumab treatment achieved longer OS and PFS. Further prospective studies and long-term evaluations in large patient populations are required to identify the useful predictive markers for determining patients with mUC who should continue pembrolizumab treatment for a relatively long term.

## Figures and Tables

**Figure 1 biomedicines-10-02243-f001:**
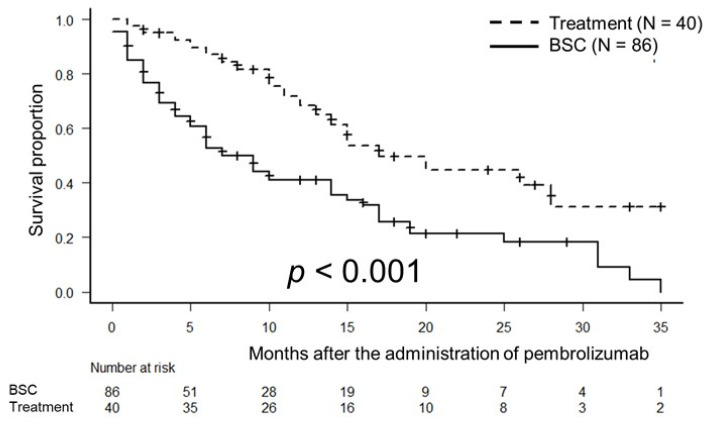
Kaplan–Meier estimates of overall survival (OS) in patients with metastatic urothelial carcinoma who received third-line therapy (treatment group) and those who did not receive third-line therapy (BSC group). The median OS times were 17 months in the treatment group and 9 months in the BSC group (*p* < 0.001).

**Figure 2 biomedicines-10-02243-f002:**
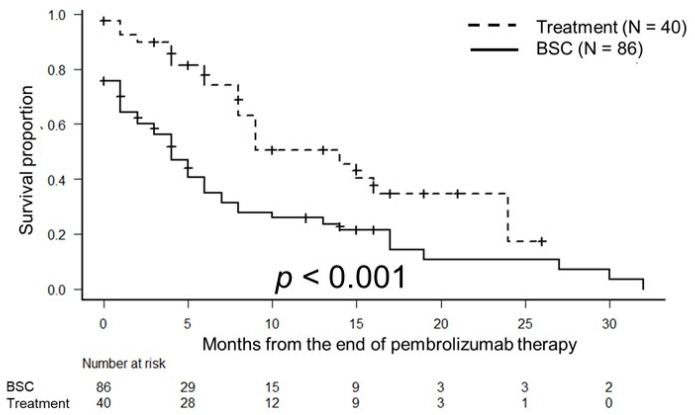
Kaplan–Meier estimates of progression-free survival (PFS) in patients with metastatic urothelial carcinoma who received third-line therapy (treatment group) and those who did not receive third-line therapy (BSC group). The median PFS times were 14 months in the treatment group and 4 months in the BSC group (*p* < 0.001).

**Figure 3 biomedicines-10-02243-f003:**
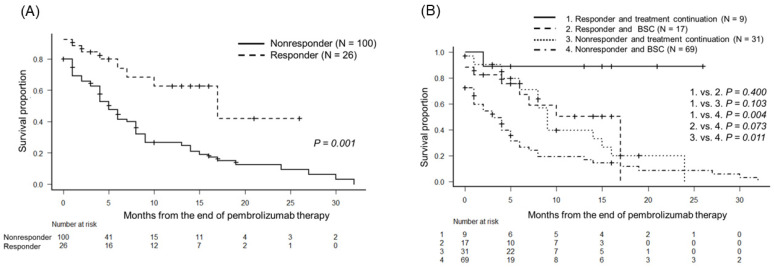
Kaplan–Meier estimates of overall survival (OS) in patients with metastatic urothelial carcinoma who showed tumor shrinkage (responder group) and who did not show tumor shrinkage (nonresponder group) after pembrolizumab administration. (**A**) The median OS times were 5 months and 17 months in the nonresponder and responder groups, respectively (*p* = 0.001). (**B**) The enrolled patients were divided into four groups: the patients who received and did not receive third-line treatment in the responder group (groups 1 and 2, respectively) and in the nonresponder group (groups 3 and 4, respectively). The median OS times were as follows: not reached in group 1, 17 months in group 2, 9 months in group 3, and 3 months in group 4. The OS of patients in the responder group treated with third-line therapy was significantly longer than that of patients in the nonresponder group not treated with third-line therapy (*p* = 0.004). In the nonresponder group, patients who received third-line treatments had significantly longer OS than those who did not receive third-line treatments (*p* = 0.011).

**Table 1 biomedicines-10-02243-t001:** Patient clinical characteristics.

Covariates	BSC Group	Treatment Group	*p*
Number	86	40	
Follow up period (median, months, interquartile range)	6.0 (2.2–13.7)	12.5 (7.7–18.5)	<0.001
Age (median, year, interquartile range)	73.5 (70.0–79.0)	71 (66.7–75.2)	0.044
Sex (number, %)			0.132
Male	59 (68.6)	33 (82.5)	
Female	27 (31.4)	7 (17.5)	
Body mass index (median, kg/m^2^, interquartile range)	21.3 (19.3–24.1)	22.6 (19.3–25.2)	0.279
The Eastern Cooperative Oncology Group performance status (number, %)			
0	1 (5.6)	9 (26.5)	0.008
1	3 (16.7)	15 (44.1)	
2	7 (38.9)	7 (20.6)	
3	5 (27.8)	3 (8.8)	
4	2 (11.1)	0 (0.0)	
Primary site (number, %)			0.285
Bladder	49 (57.0)	24 (60.0)	
Upper urinary tract	27 (31.4)	8 (20.0)	
Bladder and Upper urinary tract	10 (11.6)	8 (20.0)	
Definitive therapy for primary site (number, %)	73 (84.9)	35 (87.5)	0.625
Location of metastases (number, %)			
Lung	40 (46.5)	20 (50.0)	0.848
Liver	19 (22.1)	7 (17.5)	0.641
Bone	17 (19.8)	9 (22.5)	0.814
Lymph node	66 (76.7)	29 (72.5)	0.659
Best overall response after pembrolizumab therapy (number, %)			
Complete response	1 (1.2)	4 (10.0)	0.128
Partial response	16 (18.6)	5 (12.5)	
Stable disease	18 (20.9)	9 (22.5)	
Progression disease	51 (59.3)	22 (55.0)	

**Table 2 biomedicines-10-02243-t002:** Clinical covariates before and after pembrolizumab.

Covariates	BSC Group	Treatment Group	*p*
Number	83	40	
Before pembrolizumab therapy			
Hemoglobin (median, g/dL, interquartile range)	10.2 (8.8–11.4)	11.1 (10.2–12.6)	0.005
Albumin (median, g/dL, interquartile range)	3.4 (3.1–3.9)	3.9 (3.5–4.1)	<0.001
C-reactive protein (median, mg/L, interquartile range)	1.50 (0.50–4.52)	0.24 (0.10–0.90)	<0.001
Neutrophil-to-lymphocyte ratio (median, interquartile range)	3.29 (2.29–6.00)	2.52 (1.61–3.21)	<0.001
After pembrolizumab therapy			
Hemoglobin (median, g/dL, interquartile range)	9.8 (8.5–10.7)	11.3 (10.0–13.4)	0.041
Albumin (median, g/dL, interquartile range)	2.8 (2.5–3.6)	3.6 (3.3–4.0)	0.003
C-reactive protein (median, mg/L, interquartile range)	4.81 (1.33–10.24)	0.66 (0.14–4.34)	0.007
Neutrophil-to-lymphocyte ratio (median, interquartile range)	6.80 (3.91–8.58)	3.46 (2.21–4.88)	0.003

**Table 3 biomedicines-10-02243-t003:** Univariate and multivariate Cox proportional hazard regression analysis according to overall survival.

Covariates	Univariate				Multivariate	
	HR	95% CI	*p*	HR	95% CI	*p*
ECOG-PS, ≥2/≤1	5.859	2.506–13.7	<0.001	5.031	1.179–21.470	0.029
Lung metastasis, With/Without	1.581	1.007–2.484	0.047			
Liver metastasis, With/Without	1.752	1.058–2.903	0.029	4.281	1.192–15.370	0.026
Hemoglobin (g/dL), ≥10.8/<10.8	0.364	0.160–0.731	0.016			
Albumin (g/dL), ≥3.5/<3.5	0.275	0.122–0.621	0.002			
C-reactive protein (mg/L), ≥0.54/<0.54	6.787	2.033–22.660	0.002			
Neutrophil-to-lymphocyte ratio, ≥3.91/<3.91	12.280	4.046–37.270	<0.001			
Best response after pembrolizumab therapy, Responder/nonresponder	0.352	0.175–0.707	0.003			
Third-line therapy, With/Without	0.434	0.258–0.732	0.002			

HR, hazard ratio; CI, confidence interval; ECOG-PS, Eastern Cooperative Oncology Group Performance Status.

## Data Availability

Requests for the data presented in this study should be addressed to the corresponding author. The data are not publicly available for privacy and ethics reasons.

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
