# Peer review of "Oncological Outcomes in Patients with Metastatic Urothelial Carcinoma after Discontinuing Pembrolizumab as a Second-Line Treatment: A Retrospective Multicenter Real-World Cohort Study"

_biomedicines, 2022, doi:10.3390/biomedicines10092243_

Round 1

Reviewer 1 Report

Author in the review article summarize the oncological outcomes in real-world daily clinical practice after platinum-based chemotherapy followed by pembrolizumab for mUC. Earlier this Bellmunt and group demonstrated the same study in 542 patients entitled “Pembrolizumab as Second-Line Therapy for Advanced Urothelial Carcinoma” in 2017. How the current study is different from the Bellmunt study. In this study pembrolizumab at a dose of 200 mg every 3 weeks was used.  The author in the manuscript did not mention any dose with respect of BMI. Considering the fact that this patients are of high grade cancer then, how did author normalize the affect of ongoing other therapies in patients to see the affect of pembrolizumab. The data present in the manuscript is only of 40 patients who went through  treatments as third-line therapy. 

Author Response

Responses to the reviewer's comments

We would like to thank the Reviewers for taking the time and effort necessary to review the manuscript. We sincerely appreciate all the valuable comments and suggestions, which helped us to improve the quality of the manuscript.

Response to Reviewer 1

The authors appreciate the Academic Editor’s comments. The authors’ point-by-point responses to the comments are given below.

1) Earlier this Bellmunt and group demonstrated the same study in 542 patients entitled “Pembrolizumab as Second-Line Therapy for Advanced Urothelial Carcinoma” in 2017. How the current study is different from the Bellmunt study.

Response:

The authors have added the following sentences on line 207:

In the KEYNOTE045 trial, Bellmunt et al. demonstrated the efficacy of pembrolizumab in patients with advanced UC who experienced disease relapse or progression after first-line platinum-based chemotherapy [9]. Although oncological outcomes in patients with advanced UC or mUC receiving pembrolizumab as second-line therapy were evaluated in the trial, the efficacy of third-line therapy after pembrolizumab discontinuation was not investigated [9]. Therefore, we focused on patients with mUC who discontinued pembrolizumab in our study.

2) In this study pembrolizumab at a dose of 200 mg every 3 weeks was used.  The author in the manuscript did not mention any dose with respect of BMI.

Response:

The authors have added the following sentences on line 99:

In the KEYNOTE045 trial, pembrolizumab was administered at a dosage of 200 mg every three weeks. Therefore, in Japan, the treatment dose of pembrolizumab is 200 mg every three weeks, regardless of the BMI of the patient; the enrolled patients in this study also received 200 mg pembrolizumab every three weeks.

3) Considering the fact that this patients are of high grade cancer then, how did author normalize the affect of ongoing other therapies in patients to see the affect of pembrolizumab.

Response:

The authors have added the following sentence on line 255:

In our study, patients who achieved CR or PR after pembrolizumab administration had better oncological outcomes than those with SD or PD. This result suggests that patients with CR or PD might have maintained good ECOG-PS and better OS after receiving third-line chemotherapy.

4) The data present in the manuscript is only of 40 patients who went through treatments as third-line therapy.

Response:

The authors have added the following sentence on line 275:

Additionally, approximately 70% of the patients in this study were unable to receive third-line chemotherapy.

Reviewer 2 Report

Congratulations for such an interesting study! However, there are some aspects that are not very concise: the exact inclusion criteria were not defined. The discussions do not present a real debate between the outcomes from the studies and the literature. The Conclusions are too general and there is a lot of unuseful text.

Author Response

Responses to the reviewer's comments

We would like to thank the Reviewers for taking the time and effort necessary to review the manuscript. We sincerely appreciate all the valuable comments and suggestions, which helped us to improve the quality of the manuscript.

Response to Reviewer 2

The authors appreciate the Academic Editor’s comments. The authors’ point-by-point responses to the comments are given below.

1) There are some aspects that are not very concise: the exact inclusion criteria were not defined.

Response:

The authors have added the following sentences on line 85:

None of the enrolled patients received immunotherapy as first-line therapy. We excluded patients whose treatment response was not evaluated after pembrolizumab initiation and those with missing data.

2) The discussions do not present a real debate between the outcomes from the studies and the literature.

Response 1:

The authors have added the following sentences on line 229:

In our study, patients who received third-line therapy had a longer OS (17 months) than those in a previous study [18]. Additionally, a relatively large proportion of patients with a short duration of anticancer agent use might have enrolled in our study owing to early recurrence or metastases after chemotherapy. Therefore, patients who undergo third-line chemotherapy may have relatively improved sensitivity to anticancer drugs. Hence, these patients might achieve improved oncological outcomes after third-line therapy.

The authors have added the following sentences on line 255:

In our study, patients who achieved CR or PR after pembrolizumab administration had better oncological outcomes than those with SD or PD. This result suggests that patients with CR or PD might have maintained good ECOG-PS and better OS after receiving third-line chemotherapy.

3) The Conclusions are too general and there is a lot of unuseful text.

Response 2:

The authors have deleted the following sentence on line 281:

Additionally, this study suggested that the patients who had better clinical response to pembrolizumab treatment may achieve longer OS and PFS.

Round 2

Reviewer 1 Report

The author address the concerns and comments 

Reviewer 2 Report

The manuscript is suitable for publication after this revision.